# A Cross-Sectional Study of the Satisfaction with, Adherence to, and Perspectives toward COVID-19 Preventive Measures among Public Health Students in Jazan, Saudi Arabia

**DOI:** 10.3390/ijerph19020802

**Published:** 2022-01-12

**Authors:** Mohammed J. Almalki

**Affiliations:** Department of Health Services Management, College of Public Health & Tropical Medicine, Jazan University, Jazan 45142, Saudi Arabia; mjalmalki@jazanu.edu.sa; Tel.: +966-17-329-5000

**Keywords:** public health students, COVID-19 preventive measures, satisfaction, adherence, perspectives, Jazan, Saudi Arabia

## Abstract

The COVID-19 pandemic has had a significant influence on education systems, educational environments, teachers, and students. This study aims to assess the satisfaction with, adherence to, and perspectives toward COVID-19 preventive measures among public health students in Jazan, Saudi Arabia in order to enhance their campus experience. This study utilized a cross-sectional methodology. Data were collected using convenience sampling between 2–19 November 2020; this was accomplished using an online survey administered via Google Forms. The final sample consisted of 200 participants. More than half of the participants (55.0%) were dissatisfied with the preventive measures that had been applied on campus, while 19.0% had a neutral outlook. Interestingly, female participants showed a higher level of dissatisfaction toward the preventive measures (70.6%) than male participants (43.5%). The ordinal logistic regression analysis revealed a significant correlation between the degree of satisfaction with the COVID-19 preventive measures and the participant’s gender and education level. Male students were more likely to be satisfied with the preventive measures (*p* < 0.029, OR = 2.199) than female students. In addition, final year students were 4.1 times more likely to be satisfied with the COVID-19 preventive measures (*p* < 0.004, OR = 0.242) than Year 2 students, and 6.2 times more likely to be satisfied (*p* ≤ 0.001, OR = 0.162) than Year 3 students. Efforts are needed to improve the students’ satisfaction with COVID-19 preventive measures. Steps are also required to ensure that the procedures and actions introduced by the college extend to all students. In addition, further research is needed to recognize and understand the participants’ experiences after moving to the new buildings, receiving COVID-19 vaccines, and returning to on campus study.

## 1. Introduction

The COVID-19 pandemic has not only affected people’s health; it has influenced almost all other aspects of life, including education systems, educational environments, teachers, and students. One of the main challenges faced by the traditional education systems during the COVID-19 pandemic was adapting the system to meet the education needs [1]. Similar to other countries, Saudi Arabia was not immune to these effects and challenges. The number of people infected with COVID-19 in Saudi Arabia as of November 2020, the time of data collection was as high as 357,360, in addition to approximately 5896 deaths [2]. Current statistics by World Health Organization (WHO) revealed that the confirmed infected cases of COVID-19 by December 2021 reached 551,462, with 8867 deaths [3]. Saudi Arabia has spent a significant amount of effort and taken several measures to control the COVID-19 pandemic and limit its spread among the general public. These preventive measures included an internal curfew, public gatherings prohibition, and a ban on travel to and from Saudi Arabia. In addition, most government departments, malls, and mosques were closed. Visit, tourist, and Hajj visas were also temporarily suspended. Further, the government authorities launched national health awareness and vaccination campaigns [4,5,6,7]. Among the crucial efforts was the introduction of online education as the primary mode of teaching and learning in Saudi Arabia for both public schools and universities [8,9]. Universities in Saudi Arabia are led and supervised by the Ministry of Education through the University Affairs Council [10]. While universities are subject to national education and health policies, they retain greater autonomy in their decisions than public schools. For example, some universities insisted that students take exams on campus, despite the implementation of online education during the COVID-19 pandemic [11]. Some disciplines also required students to attend clinical and practical lessons held on campus. If preventive measures are not applied correctly, students may contract the disease or unintentionally transmit the virus to others [12]. Therefore, the health authorities and the Ministry of Education in Saudi Arabia have placed a significant amount of emphasis on the application of preventive measures within educational facilities to contain the COVID-19 pandemic [13]. These preventive measures on campus included maintaining social distancing, wearing face masks, avoiding shaking hands with others or touching public surfaces, washing or sanitizing hands as recommended, and checking the temperature and symptoms of COVID-19 at facility entrances. In addition, students were encouraged to receive COVID-19 vaccines, which were provided by universities and health authorities free of charge [14].

University students comprise a large segment of Saudi society (13.4%) [15] and are exposed to behaviors that may affect their health and spread infectious diseases, such as COVID-19. The transmission of infection between students is a danger to their families and society. Additionally, university students in the border region of Jazan may face higher risks regarding the COVID-19 pandemic than their peers in the other Saudi regions. The densely populated Jazan region is located in southern Saudi Arabia near the Yemeni border. Its border location has facilitated the crossing of illegal immigrants from Yemen and African countries [16], which increases the risk of cross-border transmission of infectious diseases [17,18]. Therefore, in the COVID-19 era, there is a need to study the effectiveness of preventive measures applied on campus and determine the degree to which students are adhering to these measures, as well as how satisfied they are regarding these controls.

This study aims to assess the satisfaction with, adherence to, and perspectives toward COVID-19 preventive measures among public health students in Jazan, Saudi Arabia. To ensure clarity, concepts used in this study, including satisfaction, adherence, and perspective, were defined. Satisfaction refers to how on campus COVID-19 preventive measures fulfill each participant’s expectations and needs. Adherence refers to the commitment to which participants follow the recommended COVID-19 preventive measures. Finally, perspective refers to the participants’ opinions regarding the implementation of the COVID-19 preventive measures as well as any related activities.

## 2. Materials and Methods

### 2.1. Study Design and Sample

This study used a cross-sectional design and a convenience sampling technique. Ebi Info version 7.2.5 [19] was used to determine the appropriate sample size (197 students) based on a 50% response distribution, a 5% margin of error, and a 95% confidence interval. The eligibility criteria for the inclusion of participants included being a student at the College of Public Health and Tropical Medicine (CPHTM) at Jazan University, enrolled at the second year or above of undergraduate studies and attending on campus courses as well as the mid-term exams conducted during Semester 1, 2020/2021 academic year. The targeted population for this study consisted of 405 students (epidemiology = 117; health education and promotion = 162; health informatics = 126).

### 2.2. Questionnaire

The questionnaire was developed in two stages. In the first stage, a group discussion concerning the potential content of the questionnaire was conducted within the CPHTM. Following this discussion, the author developed an English version of the survey. The comprehensiveness and clarity of this version were assessed by a public health specialist as well as an English language editor. In the second stage, the author translated the English version of the questionnaire into Arabic; this was necessary as all of the participants in this study spoke Arabic.

The clarity of the translation was assured by a public health researcher who was fluent in both Arabic and English. The questionnaire was then reviewed and approved by the Research Unit at the CPHTM. The questionnaire consisted of background information and questions regarding the COVID-19 preventive measures. Background information included questions on the participant’s gender, age, their discipline of study, education level, and whether the participant had contracted COVID-19.

The second part of the questionnaire was divided into four sections: the participant’s perspective of the organization and support, their perspective of preventive measures, personal adherence, and general satisfaction. The section regarding the participant’s perspective of the organization and support provided by the institution consisted of three questions on a 5-point Likert scale ranging from strongly agree to strongly disagree. The perspective of preventive measures section consisted of eight questions on a 5-point Likert scale ranging from strongly agree to strongly disagree. In addition, there was an open-ended question about the best source of COVID-19 health education provided by the CPHTM. The personal adherence section consisted of three questions on a 5-point Likert scale ranging from strongly agree to strongly disagree. This section also contained an open-ended question regarding the main barriers to the participant’s adherence to the COVID-19 preventive measures present on campus. Finally, each participant’s level of general satisfaction was assessed with a single question on a 5-point Likert scale ranging from strongly agree to strongly disagree. Another open-ended question was added to this section to identify factors that influenced the participant’s satisfaction or dissatisfaction with the COVID-19 preventive measures in the CPHTM. For more details on the different sections of the questionnaire, please refer to Appendix A.

### 2.3. Data Collection

The data were collected between 2–19 November 2020 (Semester 1, 2020/2021 academic year) using convenience sampling administered via a Google form. The author distributed the survey through student emails, student groups on WhatsApp, and Twitter. In addition, participants were encouraged to share the survey link with their colleagues. This study was approved by the Jazan University Committee of Research Ethics (REC42/1/148). Informed consent was obtained through a question regarding the willingness of each student to participate in the study. If they consented, they could move to the next step of the questionnaire; otherwise, they were automatically forced to leave the survey.

### 2.4. Data Analysis

IBM SPSS Statistics version 27 was used to manage, clean, store, and analyze the collected data. Frequencies, percentages, and descriptive statistics were reported for each variable. Ordinal logistic regression (OLR) was used to assess the association between the general satisfaction level of participants (the dependent variable) and the participants’ gender, age, discipline of study, education level, and whether they contracted COVID-19. Several independent variables were also regrouped for OLR analysis. The variable corresponding to the level of general satisfaction was regrouped from five to three groups and given the following values: agree = 3, neutral = 2, and disagree = 1. Demographic variables with two or less categories were coded as follows: gender (male = 0, female = 1); age (≤21 year = 0, >21 year = 1); and whether the participant had contracted COVID-19 (no = 0, yes = 1). Demographics with three categories were coded as follows: discipline of study (epidemiology = 1, health education and promotion = 2, health informatics = 3); and education level (Year 2 students = 1, Year 3 students = 2, final year students = 3). The content of the open-ended questions was analyzed, and the factors identified were categorized into a selection of main factors. Each factor was coded to be 0 if it was not reported by a participant and 1 if it had been reported.

## 3. Results

### 3.1. Participant Demographics

A total of 249 students completed the survey. Forty-nine duplicates were found and discarded—these were most likely generated by Internet issues and other technical reasons. The final sample was 200 participants with a mean age of 21.56 years (SD = 1.48, range 18–24). Over half of the participants (51.5%) were aged 21 years or younger, 57.5% were male, 41.5% were from the health education and promotion department, and 40.5% were enrolled in year two at the CPHTM. Remarkably, 36.0% of the participants had contracted COVID-19 (Table 1).

### 3.2. Perspectives on Organization and Support

Table 2 shows that 59.0% of the total participants either agreed or strongly agreed that there were health measures in place that screened for COVID-19 symptoms. However, 43.0% disagreed or strongly disagreed that the students were received in an organized manner. While 39.5% either agreed or strongly agreed that they knew how to act and to whom to refer to for support when they faced a concern related to COVID-19, 43.5% either disagreed or strongly disagreed with this notion, and 17.0% were neutral.

### 3.3. Participants’ Perspective of the COVID-19 Preventive Measures

The participants were asked if safe distancing measures were observed during the screening process as well as if they were generally followed on campus; 58.5% either disagreed or strongly disagreed. Furthermore, 48.5% of the participants disagreed or strongly disagreed regarding the availability of a place in which suspected COVID-19 cases could be isolated, while 26.0% were not sure. In addition, 79.0% of the participants either agreed or strongly agreed that each person’s temperature was checked immediately before entering the buildings. Furthermore, 70.5% and 62.5% of the participants agreed that students and employees used protective measures such as masks, respectively. More information is presented in Table 3.

The participants were asked about the best COVID-19 awareness-raising means on campus. Nearly one-third of the participants (29.5%) indicated that adherence by other individuals on campus to the preventive measures was the most effective mean of awareness raising. Others noted that the college had published materials on social media platforms (17.5%), had provided posters and flyers on campus (10.0%), and/or had received direct health education from the college staff (5.5%). However, 32.0% of the total participants stated that they received no health education regarding COVID-19.

### 3.4. Personal Adherence to the COVID-19 Preventive Measures

Table 4 shows that 92.5% of participants wore face masks, 76.5% washed or sanitized their hands as recommended, and 70.0% adhered to the social distancing measures.

The participants identified several barriers to their adherence to the COVID-19 preventive measures. These barriers included inadequate preventive measures (34.0%), non-adherence of some students and staff (27.0%), limited campus space (19.5%), misconceptions (14.0%), and the lack of experience regarding expected behavior during a pandemic (8.5%). Interestingly, a few participants (2.5%) mentioned the climate conditions as the main reason, with one participant noting that “It is hard to wear a face mask in this hot climate”.

### 3.5. General Satisfaction with the COVID-19 Preventive Measures

More than half of the participants (55.0%) were dissatisfied with the COVID-19 preventive measures applied within the campus, while 19.0% were neutral. In addition, it was observed that more female participants (70.6%) were dissatisfied with the measures compared to 43.5% of male participants (Figure 1).

The students listed the following as some of the perceived causes of dissatisfaction with the COVID-19 preventive measures: a lack of physical distancing, the obligatory of holding in-person exams instead of online exams, the lack of organization, the lack of screening measures in general, and the failure to use a face mask by some students and employees. Figure 2 shows that 52.9% of female participants and 41.7% of male participants were dissatisfied with the lack of social distancing due to limited campus space. Furthermore, 35.3% of female participants were dissatisfied with screening measures in general compared to 23.5% of male participants. In addition, 20.0% of male participants and 27.1% of female participants reported a lack of organizational procedures as an important cause of their dissatisfaction. Furthermore, the fact that some people failed to use a face mask was another important reason for the dissatisfaction of 17.6% and 4.3% of female and male participants, respectively. Finally, 13.0% of male participants and 16.5% of female participants were dissatisfied with taking in-person exams during this pandemic, preferring to take online exams instead. In contrast, 18.3% of male participants reported the availability of screening measures as a source of their satisfaction, compared to only 5.9% of female participants (Figure 2).

### 3.6. Ordinal Regression Analysis

OLR analysis was performed to assess the impact of independent demographic variables on the participants’ general satisfaction with the preventive measures that had been implemented (Table 5). The independent variables for the model included gender, age, discipline, education level, and whether the participant had contracted COVID-19. The full model was found to be statistically significant, with *X*^2^ (7, *N* = 200) = 41.05, and *p* < 0.001. The model explained between 18.6% (Cox and Snell R square) and 21.5% (Nagelkerke R squared) of the variance in general satisfaction. However, only two independent variables were found to have a unique, statistically significant influence on the model: gender and education level. Male students were more likely to be satisfied with the preventive measures (*p* < 0.029, OR = 2.199) than female students. In addition, final year students (Group 3) were 4.1 times more likely to be satisfied with the COVID-19 preventive measures (*p* < 0.004, OR = 0.242) than the Year 2 students (Group 1), and 6.2 times (*p* ≤ 0.001, OR = 0.162) more likely to be satisfied than the Year 3 students (Group 2).

## 4. Discussion

Our findings revealed that more than half of the participants (55.0%) were dissatisfied with the preventive measures implemented within the college, while 19.0% were neutral. Interestingly, more female participants were dissatisfied with the preventive measures (70.6%) than male participants (43.5%). This was consistent with the significant association between the gender of participants and their satisfaction with the COVID-19 preventive measures. These findings are inconsistent with prior research, which reported that females were generally more satisfied than their male counterparts [20].

In addition, final year students were 4.1 times more likely to be satisfied with the COVID-19 preventive measures than Year 2 and Year 3 students. The higher the student’s educational level, the higher their level of satisfaction with the COVID-19 preventive measures implemented on campus. One explanation for this observation might be that higher-level students have the knowledge and skills to distinguish between proper and improper preventive practices. This suggestion is supported by prior studies from China [21] and Turkey [22]. In addition, their superior knowledge and skills might help them discern more relevant information from more reliable sources. Another possible explanation is that students in the final year had the opportunity to study in person before the emergence of COVID-19, meet professors and health practitioners, deal with the public, and visit the related authorities and institutions during field training. In contrast, the university shifted new students to virtual education before gaining the chance to develop their practical experiences. During COVID-19, most participants were worried about being infected through in-person classes and exams.

Reasons for the participants’ dissatisfaction included the lack of physical distancing, in-person classes and exams, a lack of organization, a lack of screening measures in general, and the failure of some students and employees to correctly wear face masks.

Most of the participants in our study were dissatisfied with the in-person classes and exams as they preferred online activities during COVID-19. A prior study found that students with previous experience in online learning systems were more satisfied with the online exam than students who had less experience with online learning [23]. However, other studies found the opposite, especially for students who were receiving a clinical education [24,25]. Jazan University, including the CPHTM, utilized an integrated education method (face-to-face education and online activities) even before the COVID-19 pandemic [26]. Although 59.0% of the participants agreed that there were health measures in place to screen people for COVID-19 symptoms before entering the college, 43.0% indicated that students were not received in an organized manner, while 24.0% were neutral in their responses. This perspective was a cause for dissatisfaction among 20.0% of male participants and 27.1% of female participants. Efforts to control the COVID-19 pandemic within the campus must be paired with improvements to the overall organization of these measures and student support. Only 39.5% of the participants knew how to act and who to consult within the campus for support, while 17.0% were neutral in their responses. Students must be able to communicate directly with the designated college departments and professionals if they have any concerns regarding COVID-19. A student’s inability to receive support and guidance when required may affect their ability to observe the implemented preventive measures and put them at risk of infection. Students worldwide face significant amounts of pressure and anxiety during this pandemic, and they need additional support to tackle the challenges related to COVID-19 [27,28,29].

Most participants agreed that staff (62.5%) and students (70.5%) of the CPHTM adhered to preventive measures such as wearing face masks, and 79.0% of the participants agreed that each person’s temperature was checked immediately before entering the campus. This suggests that the appropriate COVID-19 preventive measures are well-implemented. However, the participants also indicated that there was insufficient physical distancing, the absence of a designated isolation location, a lack of personal protection equipment, and a lack of information regarding COVID-19. In addition, only 33.5% of participants agreed that people were asked to be aware of hand hygiene or use face masks within the campus for protection against COVID-19 during the screening procedure. Generally, the lack of the COVID-19 preventive measures is a real challenge for many organizations, including schools and universities [30]; this may negatively impact students’ adherence toward the preventive measures.

Thus, it is vital to improve the COVID-19 preventive measures and encourage their adherence to students and college staff such that their safety is ensured. First, people should be encouraged to wear face masks correctly and consider appropriate hand hygiene practices. Additionally, social distancing within campus must be enforced by the college. Although 70.0% of the participants stressed their adherence to social distancing, 58.5% believed that others were not appropriately social distancing, and 17.5% were unsure how well it was applied. Even when students tried to follow social distancing measures, they were challenged by the limited space within the buildings and the non-adherence of some students and staff.

However, nearly one-third of the participants identified the adherence to preventive precautions by others within the campus as the most important source of awareness against COVID-19. Furthermore, it is essential to designate a location in which suspected COVID-19 cases could be isolated and appropriately dealt with [13]. According to the Public Health Authority [31], the unavailability of isolation location for respiratory cases within the school is a high-risk factor for the transmission of COVID-19 among the school community. According to the Centers for Disease Control and Prevention [32,33], individuals suspected of being infected with COVID-19 must be immediately separated from others, masked, and isolated in a campus-sponsored isolation zone or sent home for personal isolation. If symptoms are severe, they should be transferred to a medical institution for treatment [32].

Moreover, nearly 46.0% of participants stated that they did not receive personal support in terms of COVID-19 personal prevention equipment. It may be appropriate to support the most vulnerable students and staff with free necessary personal preventive equipment such as masks and gloves for personal use off-campus. Furthermore, only 45.5% of participants received information and instructions related to COVID-19 prevention. The CPHTM has published a series of awareness leaflets and instructions regarding COVID-19 through the university’s website and social media platforms [34]. However, it appears that the CPHTM has struggled with communicating this information to most of the participants. Although the awareness materials may be of high quality, their dissemination to the targeted population was likely inappropriate. Finding the best mode of communication for a particular audience is an important topic of research in health education [35]. Thus, it is recommended to employ a more practical method of communication when targeting college students.

Our participants identified misconceptions as one of the barriers to their adherence to the COVID-19 preventive measures. This is consistent with a recent Saudi cross-sectional study that found that more than two-thirds of the participants had various misconceptions regarding COVID-19 [36]. Social media platforms are the primary source for unverified and unreliable information such as personal opinions or views regarding COVID-19 [37]. Social media misconceptions and myths can influence an individuals’ mental health, causing anxiety, fear, restlessness, and apprehension [37]. Another barrier was the lack of experience in dealing with pandemics. Young people in Jazan have never been exposed to infectious disease outbreaks similar to COVID-19. Even during the outbreak of the Middle East respiratory syndrome coronavirus (MERS-CoV) in Saudi Arabia, the number of infected cases in Jazan was minimal compared to other regions of Saudi Arabia [38]. Thus, health education should be prioritized among students of Jazan, especially regarding COVID-19.

Interestingly, a small portion of the CPHTM students reported the climate conditions as the main barrier to their adherence. For example, one student wrote the following in the open-ended section of the questionnaire: “It is hard to wear a face mask in this hot weather”. The weather in Jazan tends to be hot and humid in summer [39]. In conjunction with the limited amount of space on campus, the hot climate will inevitably affect students’ adherence to the precautionary COVID-19 procedures. In July 2021 (after the data in this study had been collected), the CPHTM was moved to a new large campus that had been furnished with the latest technology and equipment [40]. This transition is expected to improve the students’ ability to socially distance and promote adherence to preventive measures.

A primary limitation of this study is the convenience sampling used due to circumstances associated with the COVID-19 pandemic as well as social distancing procedures. The sample consisted only of students who were available and willing to participate as the questionnaire was being distributed. In addition, the questionnaire was self-administered and completed independently by each participant according to their interpretations. However, to ensure the questionnaire was delivered to the largest segment of students, it was distributed through students’ official emails and internal student groups. Despite these limitations, the current study contributes significantly to COVID-19 research and provides insight into the experiences and perspectives of public health students in Jazan, Saudi Arabia. In addition to efforts by the university, the results of this study contributed to strengthening preventive measures, particularly after the relocation to new facilities.

## 5. Conclusions

This study aimed to assess public health students’ satisfaction, adherence, and perspectives toward COVID-19 preventive measures. More than half of the participants were dissatisfied with the current COVID-19 prevent measures. The primary reason for the participants’ dissatisfaction was the lack of social distancing due to the limited amount of space on campus. Other essential factors included the in-person classes and exams, a lack of organization, a lack of screening measures in general, and the failure of some students and employees to properly use a face mask. Therefore, it is recommended to strengthen the COVID-19 preventive measures and encourage their adherence by students and college staff to ensure their safety. The students’ perceived inability to receive support and guidance when needed may affect their ability to respond to concerns and put them at risk of infection. In addition, many participants were unaware of the college efforts to combat the COVID-19 pandemic. Direct communication between the college and its students is a meaningful way by which materials associated with COVID-19 prevention can be shared effectively. Further research is needed to explore the students’ experience and perspective after moving to the new buildings, receiving the COVID-19 vaccines, and returning to on campus study.

## Figures and Tables

**Figure 1 ijerph-19-00802-f001:**
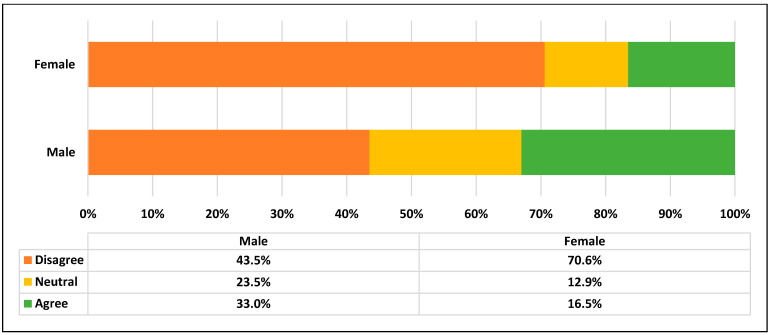
Satisfaction of the participants with the COVID-19 preventive measures by gender.

**Figure 2 ijerph-19-00802-f002:**
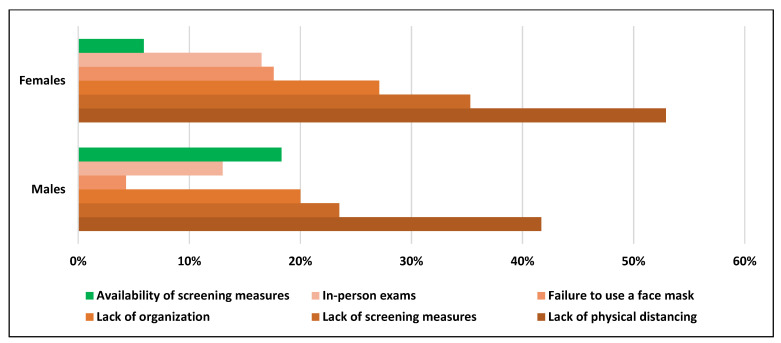
Perceived causes of satisfaction and dissatisfaction with respect to the COVID-19 preventive measures, aggregated by gender.

**Table 1 ijerph-19-00802-t001:** Sample demographic information.

Variables (*N* = 200)		*n*	%
Gender	Male	115	57.5
	Female	85	42.5
Age	≤21 years	103	51.5
	>21 years	97	48.5
Discipline of study	Epidemiology	59	29.5
	Health education and promotion	83	41.5
	Health informatics	58	29.0
Education level	Academic Year 2	81	40.5
	Academic Year 3	44	22.0
	Final Academic year	75	37.5
Contracted COVID-19	No	128	64.0
	Yes	72	36.0

Note: *N* = total number of participants who responded to this question; *n* = number of responses to each item; % = percentage of the responses to each item. Academic Year 2 is Level 3 and Level 4 equivalent, academic Year 3 is Level 5 and Level 6 equivalent, and final academic year is Level 7 and Level 8 equivalent. Level 1 and Level 2 students were enrolled at the health major preparation course. Undergraduate public health programs are four years long in addition to a one-year internship. M = male, F = female.

**Table 2 ijerph-19-00802-t002:** Participants’ perspective of the organization and support.

Questions (*N* = 200)	Strongly Agree	Agree	Neutral	Disagree	Strongly Disagree
	*n* (%)				
There are health measures to screen people for COVID-19 symptoms before entering the college facilities.	53	65	43	22	17
	(26.5)	(32.5)	(21.5)	(11.0)	(8.5)
Students are received in an organized manner.	35	30	48	49	38
	(17.5)	(15.0)	(24.0)	(24.0)	(19.0)
If there is a problem or concern related to COVID-19, I know exactly how to act and to whom within the college I can refer for support.	34	45	34	49	38
	(17.0)	(22.5)	(17.0)	(24.5)	(19.0)

Note: *N* = total number of participants who responded to this question; *n* = number of responses to each item; % = percentage of the responses to each item.

**Table 3 ijerph-19-00802-t003:** Participants’ perspective of the COVID-19 preventive measures implemented by the college.

Questions (*N* = 200)	Strongly Agree	Agree	Neutral	Disagree	Strongly Disagree
	*n* (%)				
Employees use protective measures such as face masks.	59	66	28	20	27
	(29.5)	(33.0)	(14.0)	(10.0)	(13.5)
Students use protective measures such as face masks.	80	61	32	19	8
	(40.0)	(30.5)	(16.0)	(9.5)	(4.0)
Everyone’s temperature is checked immediately before entering the buildings.	98	60	21	12	9
	(49.0)	(30.0)	(10.5)	(6.0)	(4.5)
Everyone is required to use hand sanitizer and face masks before entering the buildings.	33	34	49	46	38
	(16.5)	(17.0)	(24.5)	(23.0)	(19.0)
The safe distancing (2 m) between people is observed during the screening process and inside the campus.	24	24	35	49	68
	(12.0)	(12.0)	(17.5)	(24.5)	(34.0)
There is a designated location to isolate suspected COVID-19 cases.	23	28	52	35	62
	(11.5)	(14.0)	(26.0)	(17.5)	(31.0)
The college provides personal protective equipment for students and others, as needed.	24	39	46	43	48
	(12.0)	(19.5)	(23.0)	(21.5)	(24.0)
I have received the information and instructions related to COVID-19 preventive measures from the college.	45	46	39	30	40
	(22.5)	(23.0)	(19.5)	(15.0)	(20.0)

Note: *N* = total number of participants who responded to this question; *n* = number of responses to each item; % = percentage of the responses to each item.

**Table 4 ijerph-19-00802-t004:** Personal adherence to the COVID-19 preventive measures within the campus.

Questions (*N* = 200)	Strongly Agree	Agree	Neutral	Disagree	Strongly Disagree
	*n* (%)				
I wear personal protective equipment such as a face mask.	150	35	8	4	3
	(75.0)	(17.5)	(4.0)	(2.0)	(1.5)
I wash/sanitize my hands as recommended.	100	53	29	10	8
	(50.0)	(26.5)	(14.5)	(5.0)	(4.0)
I adhere to the social distancing measures.	82	58	33	14	13
	(41.0)	(29.0)	(16.5)	(7.0)	(6.5)

Note: *N* = total number of participants who responded to this question; *n* = number of responses to each item; % = percentage of the responses to each item.

**Table 5 ijerph-19-00802-t005:** The statistical relationship between demographics and general satisfaction level as determined using ordinal regression analysis.

Variable		B	Std. Error	Wald	df	Sig.	Odd Ratio	95% CI	
								L	U
Gender									
	0 = Male	0.788	0.360	4.782	1	0.029 *	2.199	0.082	1.494
	1 = Female	0 ^a^			0				
Education level									
	1 = Academic Year 2	−1.417	0.494	8.229	1	0.004 *	0.242	−2.385	−0.449
	2 = Academic Year 3	−1.818	0.465	15.299	1	<0.001 **	0.162	−2.730	−0.907
	3 = Final academic year	0 ^a^			0				

^a^ This parameter is set to zero because it is redundant. The dependent variable is general satisfaction. Only significant coefficients are presented in this table. * *p* < 0.05, ** *p* < 0.001.

## Data Availability

The data presented in this study are available on request from the corresponding author.

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
