# Peer review of "A Cross-Sectional Study of the Satisfaction with, Adherence to, and Perspectives toward COVID-19 Preventive Measures among Public Health Students in Jazan, Saudi Arabia"

_ijerph, 2022, doi:10.3390/ijerph19020802_

Round 1
Reviewer 1 Report
It would be interesting to know how many students enrolled in schools were in each group studied.Author Response
Please see the attachment.

Reviewer 2 Report
The authors carried out a survey on satisfaction/adherence/perspective on a convenience sample of 200 students. The study design is cross-sectional. How the authors state in the discussion section, the main limitations of this study are the small number of participants and the online modality of questionnaire transmission.
Overall, this manuscript is clear and the introduction provides a detailed picture of the issue. The methods section precisely describes the study design and the statistical analysis. Moreover, in the supplementary material are provided with the two versions of the questionnaire, which are English and original language questionnaire. Finally, the results and discussion sections show the main findings and compare them with the recent literature on this topic.
In my view, this manuscript deserves to be published in its current form.
Author Response
Thank you.
Reviewer 3 Report
This study reports results from a survey among public health students in Jazan, Saudi Arabia, regarding their perceptions of and satisfaction with COVID-19 prevention measures on campus. The author collected data from 200 students and finds significant levels of dissatisfaction with COVID-19 prevention measures. This dissatisfaction arises both from a (perceived) lack of information as well as (perceived) non-compliance of others with COVID-19 prevention measures.
This study addresses an interesting and highly relevant research question. Throughout the COVID-19 pandemic, many universities and colleges worldwide have changed their mode of operation significantly, and this has strongly affected the experience of students. At the same time, in many countries the experiences of students have not received much attention in the debate on the consequences of COVID-19 prevention policies. As such, this study reports important data. My main concern is that, while the data is potentially very interesting, the manuscript in its present form does not provide sufficient context to ensure that the findings are relevant for an international audience.
- As noted above, I would recommend that the author should provide additional information on the context of this study to ensure that the findings are relevant for an international audience. For example, international readers will likely be unfamiliar with how higher education in Saudi Arabia is organised, the demographics of the student population, or the particular characteristics of Jazan. The study also reports data that has been collected in November 2020. To better understand the information presented in the manuscript, it would be important to contextualise it by providing information on the state of the COVID-19 pandemic in November 2020 in Saudi Arabia. Also, it would be important to know which preventive measures were in place (both for the general population as well as for the student population in Jazan), and how these measures were implemented. It would also be interesting to know whether any changes with these prevention measures were implemented in response to the survey results reported in this manuscript.
- I am not convinced that it is meaningful to distinguish students by their level of education. The author uses education here essentially as a short-hand for "year of study". I would imagine this might be highly collinear with age, as in many universities students tend to start their courses at the same time and then advance at a similar pace. Moreover, the conditions can sometimes be quite different for final-year students compared to students in earlier years, e.g., regarding course load, number of exams to be taken, size of classes etc. Distinguishing between the year of study may still be interesting, but requires further information on the context to disentangle whether such differences capture differences in education, or study experience, or other factors.
